# Involvement of the Cell Division Protein DamX in the Infection Process of Bacteriophage T4

**DOI:** 10.3390/v16040487

**Published:** 2024-03-22

**Authors:** Sabrina Wenzel, Renate Hess, Dorothee Kiefer, Andreas Kuhn

**Affiliations:** Institute of Biology, University of Hohenheim, 190h, Garbenstr. 30, 70599 Stuttgart, Germany; sabrina.wenzel@uni-hohenheim.de (S.W.); r.hess@uni-hohenheim.de (R.H.); dorothee.kiefer@uni-hohenheim.de (D.K.)

**Keywords:** bacteriophage T4, DNA translocation, baseplate, tail tip, inner membrane proteins

## Abstract

The molecular mechanism of how the infecting DNA of bacteriophage T4 passes from the capsid through the bacterial cell wall and enters the cytoplasm is essentially unknown. After adsorption, the short tail fibers of the infecting phage extend from the baseplate and trigger the contraction of the tail sheath, leading to a puncturing of the outer membrane by the tail tip needle composed of the proteins gp5.4, gp5 and gp27. To explore the events that occur in the periplasm and at the inner membrane, we constructed T4 phages that have a modified gp27 in their tail tip with a His-tag. Shortly after infection with these phages, cells were chemically cross-linked and solubilized. The cross-linked products were affinity-purified on a nickel column and the co-purified proteins were identified by mass spectrometry, and we found that predominantly the inner membrane proteins DamX, SdhA and PpiD were cross-linked. The same partner proteins were identified when purified gp27 was added to *Escherichia coli* spheroplasts, suggesting a direct protein–protein interaction.

## 1. Introduction

Bacteriophage recognizes its host cell by the tail fibers contacting the bacterial outer surface, initiating the complicated process of adsorption. Myophage T4 has six long, mobile tail fibers that are involved in this first step of finding the proper host receptor in a still reversible interaction [1]. In a second step, which leads to an irreversible and tight adsorption, the short tail fibers bind to the cell surface by a concomitant movement of gp9, resulting in a conformational change within the baseplate of the phage particle [2]. The conformational change in the baseplate initiates a wave-like rearrangement of the tail sheath, resulting in a contraction of the tail [3]. Since the inner tail tube of T4 does not contract, it becomes exposed to the surrounding vicinity together with the former baseplate components gp5.4, gp5 and gp27 at its tip [2]. The stiff triple-helical structure of gp5, called the needle, is mechanically pushed through the outer membrane during the tail contraction reaction.

In the periplasm, the tip of the needle, consisting of gp5.4 and the C-terminal part of gp5 (gp5C), is thought to fall off and bind to PpiD, a periplasmic isomerase [4]. The N-terminal portion of gp5 (gp5*) encompasses a lysozyme domain that locally digests the peptidoglycan layer, paving the way for the tail tip structure to move through the periplasm towards the outer face of the cytoplasmic membrane [5]. The trimeric gp27 is bound to the N-terminal part of the gp5 trimer (gp5N) and forms a cylindrical pore suitable to allow the passage of a double-stranded DNA [6].

Currently unknown is what happens at the inner membrane. Possibly, the tail tip inserts itself into the membrane bilayer and forms a channel for the phage DNA to translocate into the cytoplasm of the host. Alternatively, the tail tip binds to an inner membrane protein that supports the phage DNA translocation. To investigate this, we have generated T4 phage particles that carry a modified gp27 protein with a His-tag. Following infection with the modified phage in the presence of a chemical cross-linker then allows to identify and, later, isolate proteins that contact gp27 during the infection process by affinity chromatography. The major gp27-bound host proteins were identified as the periplasmic isomerase PpiD, the divisome complex component DamX, and the sulfate dehydrogenase subunit SdhA.

The same cross-linking partner proteins were found when purified gp27-N-His was directly applied onto spheroplasts, suggesting that these proteins are specifically recognized by gp27. *E. coli* strains that have deletions in one of the suspected partner proteins were then tested as hosts for T4 infection. When the cell division protein DamX was deleted, the efficiency to infect a cell was significantly reduced to about 40%, suggesting that DamX considerably contributes to a highly productive T4 infection.

## 2. Materials and Methods

### 2.1. Bacterial Strains, Bacteriophages, and Culture Conditions

All bacteria, plasmids and bacteriophages used in this work are listed in Table 1. Medium preparation and bacterial growth were performed according to standard methods [7].

### 2.2. Plasmid Construction

Gene 27 of the phage T4 was amplified by PCR from the T4 genome DNA with primers that were designed for the C-terminal histidine tag to add a *Pci*I and *Mfe*I site to both ends of the PCR product. Thus, the forward primer began with a *Pci*I site as 5′-AC ATG TCA ATG TTG CAA CGC CCC GGA TAT CCA-3′ and reverse 5′-CAA TTG CTT ATT GGA AGA ACT ACT TTC TTC AGT GG-3′. For the N-terminal histidine tag, the forward primer 5′-C CAG CTC GAG ATG AGT ATG TTG CAA CGC-3′ and the reverse primer 5′-CC AGG GGA TCC TTA TTG CTT ATT GGA AGA AC-3′ were used. Each PCR was performed with GoTaq DNA polymerase (Promega) for 40 cycles with T4 DNA. The PCR products were purified (Illustra), digested for the C-terminal His-tag with *Pci*I and *Mfe*I and for the N-terminal His-tag with *Xho*I and *Bam*HI and ligated to the modified expression vectors pET22b with a TEV site and expression vector pET16b, respectively. For the propagation of the His-modified T4 *amber*27 phages, gp27-N was subcloned into pGZ119EH.

### 2.3. Purification of gp27 Constructs

*E. coli* cells were transformed with plasmids encoding gp27 with either a C-terminal 11xHis-tag (gp27-C) or an N-terminal 10xHis-tag (gp27-N). The cells were cultured in a 2xYT medium containing 200 µg/mL ampicillin at 37 °C until the cultures reached the OD_600_ of 0.6. Subsequently, the culture temperature was reduced to 18°C, and gp27 expression was induced by the addition of isopropyl-ß-D-thiogalactopyranoside (IPTG) to a final concentration of 0.5 mM. The induced cells were grown at 18 °C overnight, and harvested by centrifugation at 8000× *g* and 4 °C for 10 min the next day. The resulting cell pellets were resuspended in a lysis buffer (50 mM Tris-HCl [pH 8], 150 mM NaCl, 1 mM dithiothreitol [DTT], 1 mM ethylendiamine tetraacetic acid [EDTA]) on ice and lysed by four passages in a French pressure cell at 16,000 psi. Before cell disruption 1 mM phenylmethylsulfonyl fluoride (PMSF), a spatula tip of DNaseI and 1 mM MgCl_2_ were added. The lysate was then centrifuged at 10,000× *g* and 4 °C for 15 min. The supernatant and Ni Sepharose 6 Fast Flow slurry were incubated at 4 °C for 45 min on a rotating wheel. The sample was loaded onto a 5 mL column and the flow through collected. After washing the matrix with 20 mL of the wash buffer (50 mM Tris-HCl [pH 8], 300 mM NaCl, 35 mM imidazole), the proteins were eluted with 10 mL elution buffer (50 mM Tris-HCl [pH 8], 150 mM NaCl, 300 mM imidazole, 1 mM DTT) in 2 mL fractions. The elution fractions containing the target proteins were further purified by size-exclusion chromatography (SEC). The samples were loaded onto a Superdex 200 increase 10/300 GL column connected to an AEKTA purifier 10 system, and eluted with gel filtration buffer (20 mM Tris-HCl [pH 8], 150 mM NaCl, 10% glycerol).

### 2.4. Size Exclusion Chromatography—Multi-Angle Light Scattering (SEC-MALS)

The SEC-MALS method employed in this study was previously reported [4]. Protein samples were injected onto a Superdex 200 increase 10/300 GL column, pre-equilibrated with a SEC buffer (20 mM Tris-HCl [pH 8], 150 mM NaCl). The SEC column was integrated with a static 3-angle light-scattering detector (miniDAWN Treos II) and a refractive-index detector (Optilab T-rEX) (Wyatt Technology, Santa Barbara, CA, USA). Data acquisition was performed at a frequency of one measurement per second, maintaining a flow rate of 0.5 mL/min throughout the experiment. Data analysis was carried out using ASTRA VII software version 7.3.2.21, which facilitated the molar weight determination of the protein sample. To ensure accuracy and reliability, the light-scattering detectors underwent normalization, and data quality was verified by utilizing a bovine serum albumin (BSA) standard (Thermo, Waltham, MA, USA).

### 2.5. Liposome Preparation

The liposomes utilized in this study were prepared using *E. coli* lipids (Avanti) 1-palmitoyl-2-oleoyl-sn-glycero-3-phosphoethanolamine (POPE) and 1-palmitoyl-2-oleoyl-sn-glycero-3-phospho-glycerol (POPG). The lipids were initially dissolved in dichloromethane and subsequently evaporated under a stream of nitrogen to remove the solvent [10]. The resulting dry lipid layers were dissolved in lipid buffer (50 mM Tris-HCl [pH 7.4] and 150 mM NaCl). To mimic the membrane-lipid composition of *E. coli*, a 7:3 ratio of PE to PG was used. To obtain a homogeneous size distribution, unilamellar liposomes with an average diameter ranging from 250 to 300 nm were generated by extrusion (Mini extruder, Avanti Polar Lipids, Inc., Alabaster, AL, USA) through a 0.4 µm polycarbonate membrane (Whatman) [10,11]. Subsequently, dynamic light scattering (DLS) was used to measure the size distribution of the lipid vesicles.

### 2.6. Spheroplast Preparation

For the purpose of cross-linking experiments, spheroplasts were generated from *E. coli* C3022 cells harboring plasmid pMAL-p5X expressing the maltose binding protein MBP as periplasmic control. The bacterial culture was centrifuged at room temperature (RT) and 8000× *g* for 5 min. The cell pellet was resuspended in 50 µL spheroplasting buffer (200 mM Tris-HCl [pH 7.4], 20% sucrose, 1 mM EDTA, 30 U/µL lysozyme), and incubated for 5 min at RT. Osmotic shock was accomplished by adding 150 µL of 20 mM Tris-HCl (pH 7.4) containing 20 mM MgCl_2_ for spheroplast stabilization. The sample was further incubated for 5 min and centrifuged at 6000× *g* for 3 min to remove the spheroplasts and intact cells from the supernatant. The recovered supernatant was designated as the periplasmic fraction. The resulting pellet, containing the spheroplasts and intact cells, was then carefully resuspended in reaction buffer (20 mM Tris-HCl [pH 7.4], 150 mM NaCl, 20 mM MgCl_2_).

### 2.7. PpiD and DamX Purification

The purification protocol for PpiD has been previously documented [4]. *E. coli* PpiD was expressed from plasmid pASK-IBA3C, which encodes PpiD with a C-terminal Strep-tag (kindly provided by HG Koch, Freiburg). Cultivation was conducted in BL21 (DE3) cells in LB medium supplemented with 25 µg/mL chloramphenicol at 37 °C. Induction was initiated by adding anhydrotetracycline to a final concentration of 200 µg/L for 2 h at 37 °C. Cells were harvested via centrifugation at 10,000× *g* and 4 °C for 15 min. The resulting cell pellets were resuspended in buffer P (50 mM Tris-HCl [pH 7.4], 50 mM NaCl, 1 mM EDTA) on ice and lysed by four passages in a French pressure cell at 16,000 psi. Prior to cell disruption, 1 mM PMSF, a spatula tip of DNaseI, and 1 mM MgCl_2_ were added. The lysate was centrifugated at 35,000× *g* and 4 °C for 45 min. The pellet was resuspended in buffer A containing 1% N-lauroylsarcosine to solubilize the inner membrane at 4 °C overnight. The solubilized membrane fraction was collected by ultracentrifugation at 110,000× *g* at 4 °C for 40 min.

*E. coli* DamX, expressed from plasmid pMS119EH encoding DamX with an N-terminal Strep-tag, was cultured in MG1655 cells in LB medium supplemented with 100 µg/mL ampicillin at 37 °C until the culture reached an OD_600_ of 0.5. Induction was initiated by adding 0.5 mM IPTG for 4 h at 37 °C. Cells were harvested via centrifugation at 8000× *g* and 4 °C for 10 min. The cell pellets were resuspended in buffer D (50 mM Tris-HCl [pH 8], 150 mM NaCl) on ice and lysed by four passages in a French pressure cell at 16,000 psi. Prior to cell disruption, 1 mM PMSF, a spatula tip of DNaseI, and 1 mM MgCl_2_ were added. The lysate underwent centrifugation at 70,000× *g* and 4 °C for 30 min.

Both PpiD and DamX supernatants were loaded onto 0.5 mL of Strep-Tactin^®^ Sepharose matrix (IBA Lifescience, Göttingen, Germany). After washing the matrix with 5 mL of buffer W (100 mM Tris-HCl [pH 8], 150 mM NaCl, 0.1 mM Na_2_EDTA × 2H_2_O), the proteins were eluted with 2.5 mL of buffer E (100 mM Tris-HCl [pH 8], 150 mM NaCl, 0.1 mM Na_2_EDTA × 2H_2_O, 2.5 mM D-desthiobiotine) in 500 µL fractions. The elution fractions containing the target proteins were further purified by SEC performed with a Superdex 200 increase 10/300 GL column connected to an AEKTA purifier 10 system in buffer P for PpiD and in buffer D containing 10% glycerol for DamX.

### 2.8. Modified T4 amber27 Phage Preparation and Purification

*E. coli* C2523 cells harboring plasmid pGZ119EH encoding gp27-N-His were grown in LB medium containing 25 µg/mL chloramphenicol at 37 °C until the culture had reached an OD_600_ of 0.6. The culture was induced by the addition of 0.3 mM IPTG for 10 min. Then, the cells were infected by T4 *amber*27 phages with a m.o.i. of 5 and after 7 min superinfected by additional phage with a m.o.i. of 5 for 53 min. The cells were harvested by centrifugation with 6000× *g* at 25 °C for 15 min. The pellet was resuspended in phosphate buffer containing 1 mM MgSO_4_ at 4 °C overnight. The next day, the pellet was treated with 0.05% Triton-X-100 at RT for 1h and centrifuged at 6000× *g* for 15 min to remove the cell debris and intact cells from the supernatant. The supernatant was recovered, and the phage was further purified in a cesium chloride (CsCl) step gradient ultracentrifugation in a SW28 rotor with 110,000× *g* at 18 °C for 2 h. The phage band was collected by using a sterile syringe and cannula and dialyzed against buffer T4 (20 mM Tris-HCl [pH 7.4], 20 mM NaCl, 10 mM MgCl_2_).

### 2.9. Efficiency of T4 Phage Infection

Overnight bacterial cultures of *E. coli* strain BW25113 and its deletion mutants JW0431∆*ppiD*, JW0713∆*sdhA* and JW3351∆*damX* from the Keio collection [8] were diluted into 3 mL fresh LB medium. LB medium for the Keio collection strains included an additional 25 µg/mL of kanamycin. The cultures were grown at 37 °C until they had reached an OD_600_ of 0.6. Subsequently, the cells were infected by T4D with a m.o.i. of 5 for 1 h at 37 °C. Following the infection period, the cultures were treated with 75 µL chloroform and left for 5 min at RT. Then, the cell debris was pelleted by centrifugation of the supernatant (excluding chloroform) with 10,000× *g* for 2 min. The phage titer of each T4 infection was determined on BW25113 cell lawns. In this assay, BW25113 cells were grown to an OD_600_ of 1.0 and 300 µL of the culture was mixed with top agar and the serial dilutions of T4 phages. The mixture was then poured onto Hershey bottom plates and incubated overnight at 37 °C.

## 3. Results

### 3.1. Purification and Characterization of T4 gp27

T4 gp27 was purified from *E. coli* cells harboring either a plasmid, encoding a C-terminal 11xHis-tag (gp27-C) or an N-terminal 10xHis-tag version (gp27-N) of the protein. The gp27 protein was overexpressed from cells that were grown overnight at 18 °C and purified by nickel affinity chromatography (Ni-NTA), followed by size-exclusion chromatography (SEC). SDS-PAGE analysis of the purified proteins revealed a dominant band corresponding to the gp27 monomer (Figure 1A). The apparent molecular weight (MW) of these monomeric forms was estimated to be about 48 kDa and 45 kDa, respectively. Additionally, size-exclusion chromatography and multi-angle light scattering (SEC-MALS) were employed to analyze the stoichiometry of the purified proteins. Both gp27-C and gp27-N, eluted in a single peak with a retention time of 28.47 min and 28.08 min (Figure 1B), respectively. The molecular weights were determined to be 48.9 kDa for gp27-C and 45.6 kDa for gp27-N (Figure 1C).

### 3.2. Gp27-N Interacts Weakly with Liposomes

In the phage tail, gp27 constitutes a channel in the infecting phage particle that is conductive for the passage of phage DNA, suggesting its potential involvement of DNA translocation across the inner membrane during the infection process. To assess the binding affinity of purified gp27 proteins to a membrane, a liposome co-sedimentation assay was performed. The liposomes were generated by extrusion of a lipid mixture comprising 70% 1-palmitoyl-2-oleoyl-sn-glycero-3-phospho-ethanolamine (POPE) and 30% 1-palmitoyl-2-oleoyl-sn-glycero-3-phospho-glycerol (POPG), resulting in unilamellar liposomes [11] with an average diameter of 250 nm. The purified gp27 proteins were incubated in lipid buffer without or with liposomes at room temperature (RT) for 20 min, and centrifuged. The supernatant and pellet fractions were analyzed by SDS-PAGE and stained with Coomassie Blue (Figure 2A,B). Gp27-C remained in the supernatant (lane 3), whereas small amounts of purified gp27-N were found in association with the liposomes (Figure 2B, lane 6). As a control, the membrane associated SecA protein was used, demonstrating its binding affinity to the liposomes under the same conditions (lanes 1 and 2). Taken together, the data show a weak binding of gp27-N, whereas gp27-C did not bind to liposomes.

### 3.3. Identification of Potential E. coli Binding Partners of Purified Gp27-N

The binding of gp27 to the inner membrane could also be mediated by integral membrane proteins. To identify potential partner proteins of the inner membrane, the purified gp27-N was cross-linked with 0.44% paraformaldehyde to *E. coli* spheroplasts at RT for 20 min. The cells were lysed by 1% Triton-X-100 and the supernatant was analyzed after TALON Superflow affinity chromatography by SDS-PAGE (Figure 3A). As a control, a sample of the eluted fraction was heated to 95 °C for 15 min to reverse the cross-links (lane 2). Two major, specifically gp27 cross-linked, bands appeared with a molecular mass of 75 kDa and 150 kDa, respectively (lane 1, X-links, indicated by asterisks). Most of the purified gp27-N was not cross-linked and remained as a monomer with a molecular weight of 45.6 kDa. The two distinct gel-shifted protein bands (indicated by asterisks) were subjected to matrix-assisted laser desorption/-ionization (MALDI) for further analysis (Figure 3B). The MALDI data identified peptide sequences of the periplasmic *E. coli* chaperon PpiD, the cell division protein DamX, and the succinate dehydrogenase SdhA. They were found in the 150 kDa band as cross-links but also in the 75 kDa band, which likely contains the proteins that co-eluted with gp27 during affinity purification. The 150 kDa band also contained some peptides of AceE (100 kDa), DnaK (69 kDa), EptC (67 kDa), FtsH (71 kDa), FtsI (64 kDa), LpdA (51 kDa), NuoG (100 kDa) and SecD (67 kDa), which might be further potential partner proteins for gp27. Since the molecular weights of most of these proteins do not add up with gp27, the conclusion that they are cross-linked is only tentative and must be verified with further experimental data.

To corroborate these interactions directly without using cross-linking, DamX and PpiD were strep-purified and incubated with purified gp27-N-His for 15 min at 37 °C. The co-elution after a nickel affinity chromatography was followed by SDS-PAGE (Figure 4). In fact, considerable amounts of DamX (Figure 4A, lane 3), as well as a smaller amount of PpiD (Figure 4B, lane 3), were detected in the elution fraction together with gp27-N. This co-elution contrasts the control results when DamX (Figure 4A, lane 4) or PpiD (Figure 4B, lane 4) were incubated alone. Here, DamX as well as PpiD, respectively, were exclusively found in the unbound (flow-through) fraction.

### 3.4. Infectivity of T4 with a Complemented gp27-His

To assess whether the His-tagged gp27 constructs can be incorporated into phage progeny and complement a T4 *amber*27 phage mutant in vivo, bacterial cells containing the respective pET plasmids encoding either gp27-C or gp27-N were grown to an OD_600_ of 1.0, mixed with top agar, and poured onto Hershey plates. After solidification of the agar, serial dilutions of the T4 *amber*27 phage were spotted (5 µL) onto the plates and incubated overnight at 37 °C (Figure 5). The T4 *amber*27 phage used contain an *amber* stop codon at the beginning of gene 27. Thus, infection of a non-suppressor strain such as BL21 (DE3) is non-productive and only a small rate of the mutants can form plaques that had reverted to wild type (row a). This result is consistent with observations in BL21 (DE3) strains carrying the empty pET16b-plasmid (row c). For the determination of the phage titer of the phage stock solution, a control with the *amber* suppressor strain *E. coli* CR63 was performed (row b). The T4 *amber*27 phage generated plaques on *E. coli* BL21(DE3) harboring plasmids with gp27-C (row d) and gp27-N (row e) in quantities indistinguishable from the phage titer determined on the *amber* suppressor strain *E. coli* CR63. We conclude that both gp27-C and gp27-N can fully complement the *amber* mutant and the His-tagged proteins are, therefore, efficiently incorporated into the progeny phage. We confirmed that the DNA of these plaques has the original amber mutation by sequencing.

### 3.5. Identification of Potential E. coli Binding Partners with the Modified T4 Phage 

To perform cross-linking experiments in vivo, functional gp27 His-tagged T4 *amber*27 phage particles were generated and purified. *E. coli* cells harboring a plasmid with gp27-N were superinfected with T4 *amber*27 phage and grown for 1 h. The phage particles were purified from the cell lysate by a cesium chloride (CsCl) gradient in an ultracentrifuge and the band containing the phage was collected and dialyzed. The phage lysate was analyzed by SDS-PAGE and a Western blot with antiserum to the His-tag (Figure 6). T4 *amber*27 phage, modified with a gp27 carrying an N-terminal His-tag, assembled into virions displaying the tag of choice, as verified by Western Blots of the CsCl-purified phages (Figure 6B).

Potential transient interactions with a periplasmic component were investigated in vivo using a chemical cross-linking treatment. *E. coli* C3022 were infected by the purified and modified T4 *amber*27 phages (Figure 6A) containing gp27-N-His proteins for 10 min in the presence of 0.44% paraformaldehyde at 37 °C. The cells were lysed by 1% Triton-X-100 and 2.5 M urea, and the soluble extract was subjected to nickel affinity chromatography (Figure 7A). The formaldehyde treatment yielded three higher molecular weight bands at approximately 75 kDa, 100 kDa and 150 kDa, respectively (Figure 7A, X-links, indicated by asterisks), which were subsequently subjected to matrix-assisted laser desorption/-ionization (MALDI) for further identification (Figure 7B). This analysis showed several peptides of gp27-N, the PpiD, the DamX and the SdhA proteins, as cross-link products with a protein identification probability of 100%, respectively. The major band at around 75 kDa contained SdhA, PpiD and DamX. However, DamX was also found in the 150 kDa band, while PpiD was present in the 100 kDa band. Notably, the 150 kDa band also contained additional peptides from AceE, DnaK, FtsH, LpdA, and NuoG, consistent with the observations from the in vitro cross-link experiment of purified gp27-N with *E. coli* spheroplasts (3.3).

This result is consistent with the observation of the in vitro cross-linking experiment with purified gp27-N (Figure 3) and suggests that DamX, PpiD and SdhA are likely to play supportive roles in the T4 infection process. To get more evidence for the involvement of these three proteins in promoting the infection process, the average phage yield of T4 after a single infection cycle was analyzed in various deletion strains of the Keio collection [8]. The *ppiD* deletion strain JW0431∆*ppiD*, the *sdhA* deletion strain JW0713∆*sdhA* and the *damX* deletion strain JW3351∆*damX* were compared to the parental BW25113 with respect to their efficiency of propagation of T4 phage (Figure 8). The average yield after a one-step infection of T4 in JW0431∆*ppiD* and T4 JW0713∆*sdhA* was reduced to 71% and 70%, respectively, compared to the parental strain BW25113. Moreover, the T4 infection in the JW3351∆*damX* deletion strain resulted in a substantially reduced burst size to only about 40%. This result is an indication that T4 phage infection is less efficient when one of these proteins is missing. All three mutants were not affected in their cell division and showed normal growth.

## 4. Discussion

The bacterial host proteins that are first contacted by the infecting T4 phage are the receptor proteins in the outer membrane [12]; in addition, there is binding to the lipopolysaccharides. During the penetration of the contracted phage tail into the host cell wall, the invading tail core structure locally digests the peptidoglycan layer due to the enzymatic action of the lysozyme moiety of gp5* on the tail tip. Later in infection, the gp5 lysozyme is inactivated by the T4 spackle protein which is secreted into the periplasm [13]. The needle structure composed of gp5C and gp5.4 then has to be removed from the tail since its tight ß-helical structure would not allow the phage DNA to pass through the phage tail. This leaves gp5N and gp27 exposed at the tail tip to then interact with the membrane surface. It has been speculated [14] that gp5N also leaves the tail tip when it arrives in the periplasm and that the gp27 trimer opens up to let the DNA pass through. This might happen after contacting the surface of the inner membrane, possibly at a receptor protein. Whereas gp27 does not firmly bind to liposomes (Figure 2B; [4]), we found that it interacts with spheroplasts (Figure 3A). Based on this observation, we conclude that the incoming phage must recognize some host protein in the inner membrane or at the periplasmic side of the inner membrane.

To follow and purify T4 gp27, a His-tag was added to either the C-terminus or N-terminus, respectively. Both modified proteins were well-expressed and functional as they complemented an T4 *amber*27 infection (Figure 5d,e). We noticed that the titer of the complemented phage with gp27-C was slightly reduced compared to gp27-N. This might be caused by an interference of the His-tag with the binding of gp5N which occurs at the C-terminal region of gp27 [6].

The strategy to find such a host protein factor was to first construct a phage that had a His-tag in one of its tail tip proteins that could then serve as bait. Chemical cross-linking should hold on to candidate host membrane proteins that will be fished out by His-tag affinity chromatography. Using this approach, we identified DamX, SdhA and PpiD of *E. coli* (Figure 7B). DamX is a component of the *E. coli* divisome dedicated to organizing the septum initiation and division of the bacterial cell into daughter cells. It assembles into division ring structures which dissemble after cell division, but the protein remains in the inner membrane [15]. It also interacts with DedD and is involved in peptidoglycan synthesis [16,17]. The succinate dehydrogenase SdhA is part of the aerobic respiratory chain complex II and couples it to the TCA cycle in *E. coli* [18]. PpiD is a periplasmic chaperone with an isomerase domain involved in quality control [19]. It is anchored in the inner membrane and is found in complexes with the SecYEG translocon [20]. All these multicomponent inner membrane complexes could, in principle, act as a translocation device for the incoming phage DNA.

To test the biological relevance of the partner proteins for the infection process of T4, we investigated the deletion mutants from the Keio collection [8]. T4 was added to a growing culture of the different host strains with an m.o.i. of 5. After 1h post-infection, the cells were lysed and analyzed for their phage progeny. When compared to the reference strain BW25113, the burst size was severely reduced by 60% when *damX* was deleted, whereas a deletion of *sdhA* or *ppiD* led to a burst size reduction of T4 of 30%. We speculate that phage DNA might use these protein complexes to assist its passage through the inner membrane.

The interaction between gp27 and DamX was tested in more detail. Whereas gp27-N had a 10xHis-tag, a Strep-tag was added to DamX at its N-terminus. Both proteins were purified and tested for co-elution after affinity chromatography. The predicted molecular weight of DamX was approximately 46 kDa. However, our results demonstrate that the purified DamX protein runs at a slower rate than expected in SDS gel, at around 80 kDa (Figure 4A). This observation has been described previously [21] and probably corresponds to a DamX dimer. We conclude from this result that the proteins interact directly with each other. This will allow to characterize the interaction and topological flexibility on the structural level in the future, particularly because the structure of gp27 is known [6]. In addition, it would be interesting to know whether DamX and the divisome are involved in the translocation of the infecting DNA across the inner membrane. Possibly, the divisome structure and localization also provides a preferred adsorption site for T4. DamX may play an important role here as, recently, this protein has been shown to be involved in membrane constriction at the initiation site of cell division and appears to be very stable in the division ring structure throughout the whole division process [15].

## Figures and Tables

**Figure 1 viruses-16-00487-f001:**
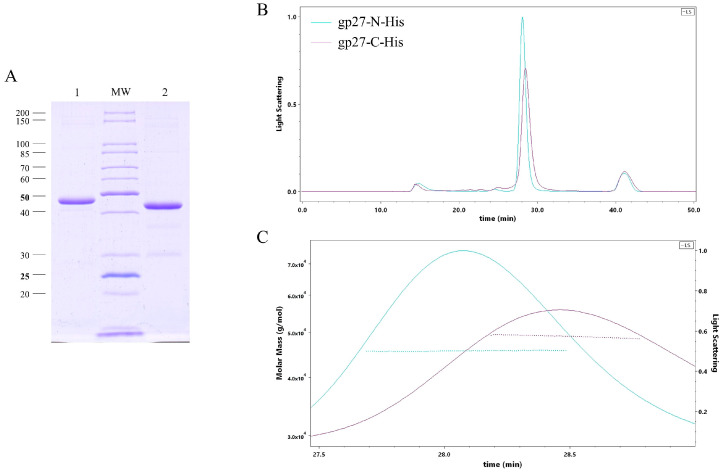
(**A**) SDS-PAGE analysis of both purified gp27 variants after nickel affinity chromatography (Ni-NTA) and gel filtration. Gp27-C-terminal 11xHis-tag (gp27-C) with an apparent molecular weight (MW) of about 48 kDa (lane 1) and gp27-N-terminal 10xHis-tag (gp27-N) with an apparent MW of about 45 kDa (lane 2). (**B**) Size-exclusion multi-angle light scattering (SEC-MALS) analysis of both gp27 variants was performed by separating affinity-purified proteins on a Superdex 200 increase column. Both gp27-C (magenta-colored) and gp27-N (cyan-colored) eluted as single peaks with retention times of 28.47 min and 28.08 min, respectively. (**C**) Calculated molar masses of 48.9 kDa and 45.6 kDa (dotted lines), respectively, corresponded to the apparent molecular weights and showed a homogeneous distribution over the entire peaks.

**Figure 2 viruses-16-00487-f002:**
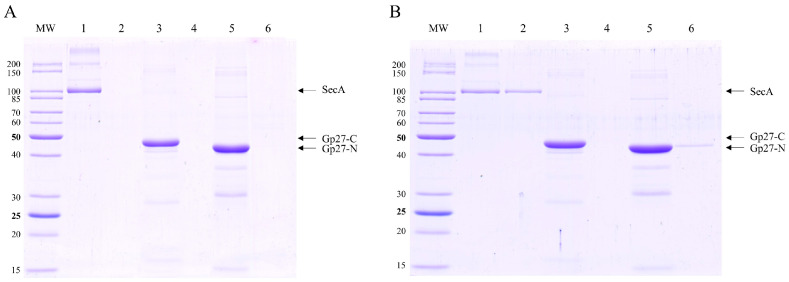
Coomassie Blue-stained SDS gels of gp27 variants incubated without and with liposomes (70% 1-palmitoyl-2-oleoyl-phosphoethanolamine [POPE] and 30% 1-palmitoyl-2-oleoyl-phosphoglycerol [POPG]) at RT and separated after centrifugation. (**A**) Binding assay without liposomes. All purified proteins were present in the supernatant (lanes 1, 3 and 5). (**B**) Liposome co-sedimentation assay. Gp27-C does not bind to liposomes (lane 3 supernatant, lane 4 pellet), whereas gp27-N binds weakly to liposomes (lane 5 supernatant, lane 6 pellet). As a control, purified SecA protein was substantially found in the pellet (lane 2).

**Figure 3 viruses-16-00487-f003:**
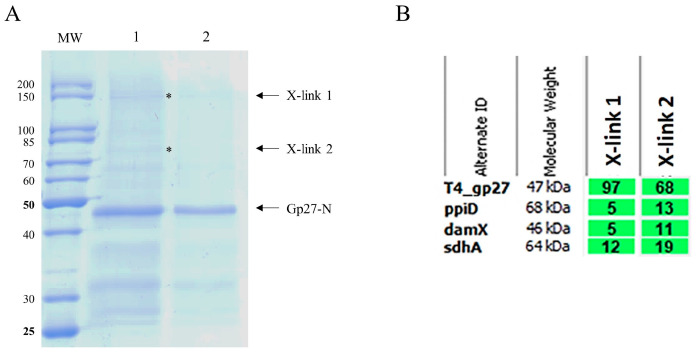
(**A**) In vitro cross-linking of purified gp27-N with *E. coli* spheroplasts. Cross-linking was performed with 0.44% paraformaldehyde for 20 min at RT, followed by nickel affinity chromatography. The elution fraction was heated to 50 °C for 5 min (lane 1) or at 95 °C for 15 min prior to SDS-PAGE to resolve the cross-link (lane 2). Cross-linked complexes are indicated by asterisks and arrows. To identify the protein bands, matrix-assisted laser-desorption/-ionization (MALDI) analysis was performed. (**B**) Evaluation of the MALDI analysis with the Scaffold 4 software version 4.8.7. Peptides of PpiD, DamX and SdhA were identified. For each identified protein, the exclusive unique peptide count is shown.

**Figure 4 viruses-16-00487-f004:**
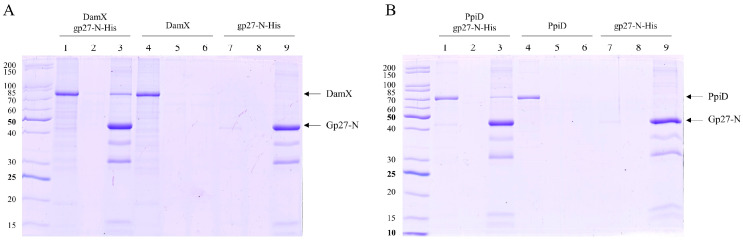
In vitro binding of gp27-N to DamX and PpiD. The purified proteins were incubated with purified gp27-N for 15 min at 37 °C, followed by affinity chromatography and SDS-PAGE. The flow-through fractions (lanes 1, 4, 7), the wash fractions (lanes 2, 5, 8) and elution fractions (lanes 3, 6, 9) are shown. (**A**) DamX substantially co-eluted with gp27-N (lane 3), whereas DamX alone was found only in the flow-through fraction (lane 4). (**B**) Analogously, PpiD co-eluted with gp27-N (lane 3), whereas PpiD alone was found only in the flow-through fraction (lane 4).

**Figure 5 viruses-16-00487-f005:**
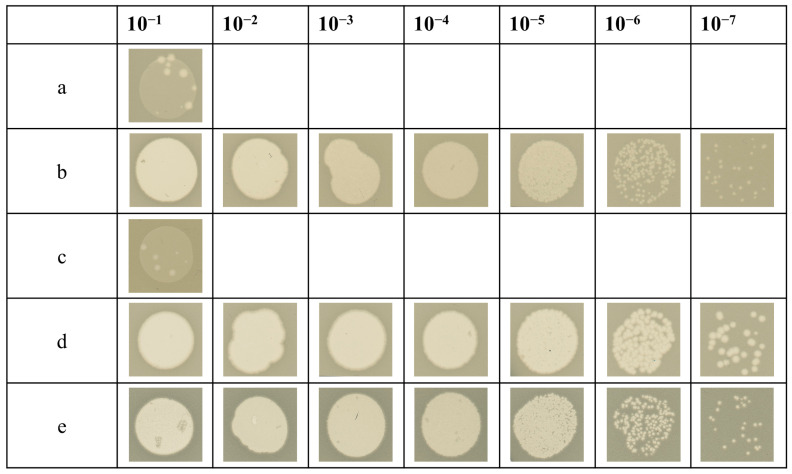
In vivo ‘spot assay’ in which serial dilutions of T4 *amber*27 phage were spotted on different bacterial lawns (**a**–**e**) to analyze the level of complementation. As controls, phages were spotted on the non-suppressor strain *E. coli* BL21 (DE3) (**a**) and the *amber* suppressor strain *E. coli* CR63 (**b**) to determine the phage titer of the solution. For complementation studies, the *E. coli* BL21 (DE3) cells harboring either an empty pET16b-plasmid (**c**) or a pET-plasmid encoding either gp27-C (**d**) or gp27-N (**e**) were used as plating bacteria. Both plasmid constructs showed plaques at about the same dilution level as *E. coli* CR63, demonstrating that both gp27-C and gp27-N expressed in the plating bacteria fully complement the *amber* mutant.

**Figure 6 viruses-16-00487-f006:**
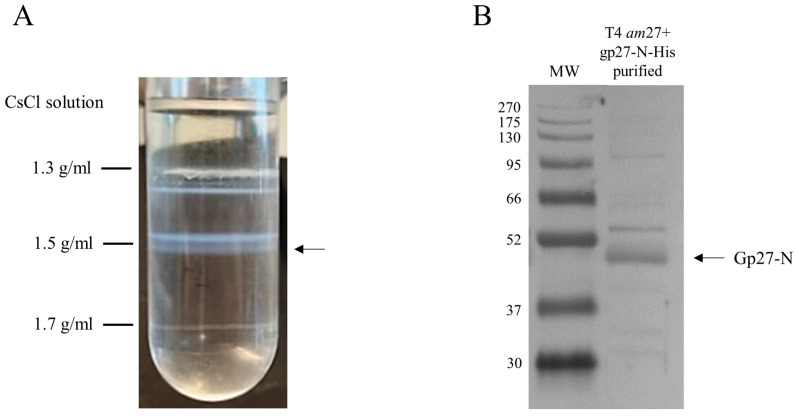
Purification of modified T4 *amber*27 phages with built-in gp27-N-His proteins. (**A**) Cesium chloride density gradient after ultracentrifugation of the gp27-N lysate from superinfected *E. coli* strain C2523, harboring plasmid pGZ119EH-gp27-N, with T4 *amber*27 phage. (**B**) Purified functional gp27 His-tagged T4 *amber*27 phage particles obtained from the CsCl-gradient in A (marked with an arrow) were analyzed by SDS-PAGE and a Western blot with His-tagantiserum.

**Figure 7 viruses-16-00487-f007:**
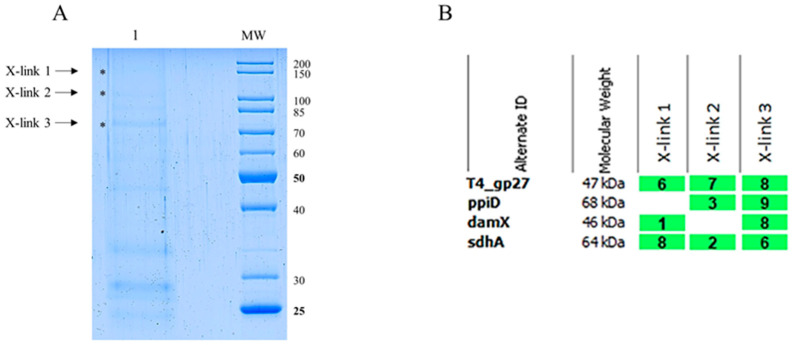
(**A**) Coomassie Blue-stained SDS gel of modified T4 *amber*27 phages with built-in gp27-N proteins in vivo cross-link experiment. *E. coli* C3022 cells were infected with gp27 His-tagged T4 *amber*27 phages and an m.o.i. of 2 in the presence of 0.44% paraformaldehyde for 10 min at 37 °C. After affinity chromatography purification of the Triton-solubilized cells, the elution fraction was heated at 50 °C for 5 min. Cross-linked complexes are indicated by asterisk and arrow. To identify the X-link product proteins, matrix-assisted laser-desorption/-ionization (MALDI) analysis was performed. (**B**) Evaluation of the MALDI analysis with the Scaffold 4 software. Peptides of PpiD, DamX and SdhA were identified. For each identified protein, the exclusive unique peptide count is shown.

**Figure 8 viruses-16-00487-f008:**
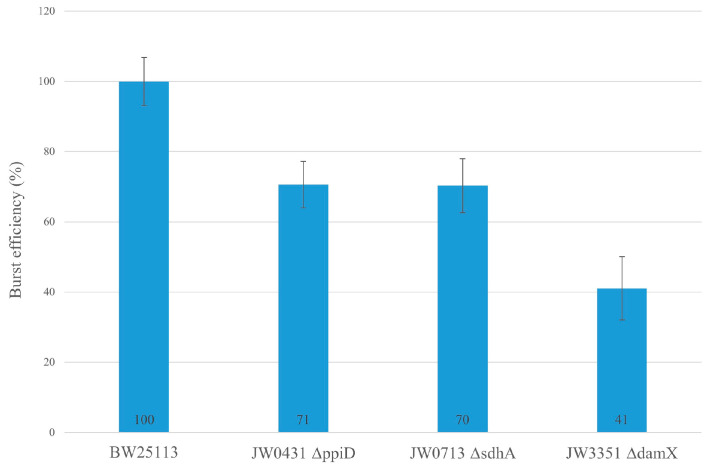
Average phage yield of T4 is affected in the *E. coli* deletion strains. JW0431 cells (∆*ppiD*), JW0713 cells (∆*sdhA*), JW3351 cells (∆*damX*) and their parental strain BW25113 were tested for their burst after T4 infection with an m.o.i. 5. The number of plaques was reduced for JW0431 to 71%, for JW0713 to 70% and for JW3351 to 41%, normalized to the average phage yield in BW25113.

**Table 1 viruses-16-00487-t001:** Bacterial strains, plasmids, and bacteriophages used in this study.

Strain, Plasmid, or Phage	Relevant Properties and Use	Source or References
*E. coli* strains BW25113 BL21 (DE3) C2523 C3022 CR63 JW0431 JW0713 JW3351 MG1655 Plasmids pASK-IBA3C pET16b_modified_ pET22b_modified_ pMAL-p5X pMS119EH pGZ119EH T4 phage T4D 27*am*N120	Parent strain of the Keio collection of single gene knockouts Carries the T7 RNA polymerase gene in lambda DE3 in the chromosome Suitable for protein expression, gp27-N protein expression T7 RNA polymerase gene in the lac operon—no lambda prophage Used for cross-linking experiments Su+ (supD): propagation of T4 *amber* mutant ∆*ppiD* Keio Collection, Kan^r^ ∆*sdhA* Keio Collection, Kan^r^ ∆*damX* Keio Collection, Kan^r^ Suitable for DamX protein overexpression For *ppiD* under tetracycline promoter, Chl^r^, C-terminal Strep-tag For gp27-N under T7 promoter, Amp^r^, N-terminal 10xHis-tag For gp27-C under T7 promoter, Amp^r^, TEV-site, C-terminal 11xHis-tag For MBP expression under tac promoter, Amp^r^ For *damX* expression under tac promoter, Amp^r^, N-terminal Strep-tag For gp27-N-10xHis-tag expression under tac promoter, Cam^r^ Wild type Gene 27 *amber* mutant, position C51T (TAG codon)	Horizondiscovery [8] Our collection [9] NEB NEB Our collection Horizondiscovery [8] Horizondiscovery [8] Horizondiscovery [8] Our collection H.G. Koch This work This work NEB This work This work Our collection Our collection

## Data Availability

All materials generated for this research are available from the authors. Details of data analysis can be provided upon request.

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
