# Peer review of "Involvement of the Cell Division Protein DamX in the Infection Process of Bacteriophage T4"

_viruses, 2024, doi:10.3390/v16040487_

Round 1
Reviewer 1 Report
Comments and Suggestions for Authors
In this paper, Andreas Kuhn’s laboratory investigated the E. coli host proteins involved in the infection mechanism of bacteriophage T4. Though the structure-function aspects of most of the phage tail and baseplate proteins are well established, the dynamic aspects of the mechanism, particularly the conformational transitions and interactions with the E. coli membrane proteins that facilitate translocation of the genome into the host are unknown. The current paper identifies host proteins that might be interacting with the tail tube/baseplate hub protein gp27 which is expected to be at the tip of the tail tube after the gp5-gp5.4 needle complex is dissociated during the infection process.
The main conclusion of the paper is that gp27 interacts with three E. coli inner membrane proteins DamX, SdhA, and PpiD. The evidence came from three independent experiments: i) crosslinking of gp27 to host proteins after adding purified gp27 to spheroplasts, ii) crosslinking of gp27 in vivo during T4 infection, and iii) direct binding of purified gp27, to purified DamX or PpiD proteins. While this is an interesting insight, there are technical concerns in the experiments.
The concerns are as follows:
1) The spot test experiment described in Section 3.4 and Figure 5 is actually a “recombination” experiment, not a “complementation” experiment. In a recombination experiment, the progeny phage and plaques produced would be wild-type due to exchange of the amber mutation with the wild-type sequence on the plasmid, whereas in the complementation experiment, the progeny phage and plaques produced would retain the amber mutation and hence not form plaques on a suppressor-minus E. coli strain. Authors can confirm this by testing the plaques produced (e.g., at 10E-7 dilution), which would be wild-type and His-tag-minus, but not His-tag-plus and amber as assumed.
This section needs to be deleted as it is not what it was supposed to be.
2) In Figures 3 and 7, there are several crosslinked bands seen on the gel. However only certain bands, were subjected to MALDI analysis. How are these bands selected, not others, because there are no major bands? Are there any other host proteins found in other crosslinked bands?
Importantly, the mol. wts. of the crosslinked complexes do not add up. In Figure 3A, the mol. wts. of the crosslinked bands are ~75 kDa and 150 kDa. Neither of them add to a crosslink between gp27 (47 kDa) with DamX 46 (kDa), PpiD (68 kDa), or SdhA (64 kDa).
In Figure 7A, the size of the crosslinked band is ~75 kDa, which again does not add up to the sizes expected when crosslinked to DamX, PpiD, or SdhA.
3) Reduction in burst efficiencies (average phage yield per infection might be a better term to use) shown in the E. coli mutants are rather modest, ~40-60%. How do we know if this reduction was due to inefficient infection or because the host is not growing well or its metabolism is affected due to these mutations and hence producing reduced yield?
Since these mutations are expected to affect the infection mechanism, but not necessarily phage assembly, a better measure would be to determine the efficiency of infection/plating rather than phage yield.

Author Response
In this paper, Andreas Kuhn’s laboratory investigated the E. coli host proteins involved in the infection mechanism of bacteriophage T4. Though the structure-function aspects of most of the phage tail and baseplate proteins are well established, the dynamic aspects of the mechanism, particularly the conformational transitions and interactions with the E. coli membrane proteins that facilitate translocation of the genome into the host are unknown. The current paper identifies host proteins that might be interacting with the tail tube/baseplate hub protein gp27 which is expected to be at the tip of the tail tube after the gp5-gp5.4 needle complex is dissociated during the infection process.
The main conclusion of the paper is that gp27 interacts with three E. coli inner membrane proteins DamX, SdhA, and PpiD. The evidence came from three independent experiments: i) crosslinking of gp27 to host proteins after adding purified gp27 to spheroplasts, ii) crosslinking of gp27 in vivo during T4 infection, and iii) direct binding of purified gp27, to purified DamX or PpiD proteins. While this is an interesting insight, there are technical concerns in the experiments.
The concerns are as follows:
1) The spot test experiment described in Section 3.4 and Figure 5 is actually a “recombination” experiment, not a “complementation” experiment. In a recombination experiment, the progeny phage and plaques produced would be wild-type due to exchange of the amber mutation with the wild-type sequence on the plasmid, whereas in the complementation experiment, the progeny phage and plaques produced would retain the amber mutation and hence not form plaques on a suppressor-minus E. coli strain. Authors can confirm this by testing the plaques produced (e.g., at 10E-7 dilution), which would be wild-type and His-tag-minus, but not His-tag-plus and amber as assumed.
This section needs to be deleted as it is not what it was supposed to be.
Answer: I think we did not explain this experiment carefully enough and changed the figure legend and the text (lines 278-286). We used the T4am27 for all the spot tests at the identical concentrations. When we spotted 5 microliters on the suppressor plate we obtained about 30 plaques at the 10E-7 dilution which is consistent with the titer of 6x10E10 phage per ml. We obtained the same number of plaques on lawns where gp27 is expressed from a plasmid that has the gp27 gene. We now added a control lane with the empty plasmid in the new figure 5. We also analysed and sequenced some of the complemented plaques and they still had the original amber mutation and lacked the His-tag. In conclusion, the experiment shows complementation and not recombination. We think it is important to show that the gp27 with the His tag is indeed functional and its expression results in infectious phages.
2) In Figures 3 and 7, there are several crosslinked bands seen on the gel. However only certain bands, were subjected to MALDI analysis. How are these bands selected, not others, because there are no major bands? Are there any other host proteins found in other crosslinked bands?
Answer: We analysed only the visible bands which were larger than gp27 (45 kDa) and we only considered the bands that contained gp27. There were other peptides found in the 150 kDa band that are potentially cross-linked to gp27. We listed them in line 249-253 and 313-318.
Importantly, the mol. wts. of the crosslinked complexes do not add up. In Figure 3A, the mol. wts. of the crosslinked bands are ~75 kDa and 150 kDa. Neither of them add to a crosslink between gp27 (47 kDa) with DamX 46 (kDa), PpiD (68 kDa), or SdhA (64 kDa).
Answer: We agree with the reviewer that the 75 kDa band may contains the non-cross-linked proteins PpiD and SdhA. We also want to point out that the cut bands have naturally a range of MWs which in this case is from about 60 to 80 kDa. In addition, the cross-linked bands could also contain additional small components that affect their apparent MWs. Cross-linked products almost never are fully unfolded in SDS-PAGE without boiling the samples (the boiling would release the cross-link as shown in lane 2) and often run to positions not corresponding exactly to the sum of the mol. weights of the single cross-link partners.
In Figure 7A, the size of the crosslinked band is ~75 kDa, which again does not add up to the sizes expected when crosslinked to DamX, PpiD, or SdhA.
Answer: We agree with the reviewer and changed the text (line 250) stating that the 75 kDa band contains the proteins without the cross-link but were likely his-affinity co-purified with gp27.
3) Reduction in burst efficiencies (average phage yield per infection might be a better term to use) shown in the E. coli mutants are rather modest, ~40-60%. How do we know if this reduction was due to inefficient infection or because the host is not growing well or its metabolism is affected due to these mutations and hence producing reduced yield?
Answer: We tested all the mutant strains and they show normal growth comparable to the parental strain. In the figure legend as well as in the graph we changed the term to “average phage yield”.
Since these mutations are expected to affect the infection mechanism, but not necessarily phage assembly, a better measure would be to determine the efficiency of infection/plating rather than phage yield.
Answer: In the presented experiments we aimed to see whether a single step infection in a liquid culture is affected. However, we also analysed the EOP and the results were in the same range of reduction.

Reviewer 2 Report
Comments and Suggestions for Authors
This paper reports on the protein interactions of gp27 of phage T4 with inner-membrane proteins, DamX, SdhA and PpiD, which appear to be important for the DNA intrusion to the host bacterium. They focused on gp27, because the trimeric protein is thought to form a cylinder to facilitate phage DNA intrusion. They not only showed that the host proteins interact with gp27, but also demonstrated that the absence of the host proteins each significantly lower the infection rate of phage T4.
The paper is well organized and easy to read. The findings will be important for the understanding the mechanism of T4 DNA intrusion and important step towards full understanding of phage infection. It will be interesting to general readers. The only thing is that they did not describe their future direction of the research, which the readers may want to see. The followings are minor points which may improve the manuscript.
1. The authors claim that the liposome they used was unilameller. They should refer to any paper which experimentally showed that the liposome they made was unilameller.
2. As the gp27 structure is known, they could envisage the effect of the His-tags’ insertion to gp27-N and gp27-C, which they did not describe.
3. The authors presented the apparent MW from the experiment in the legend of Fig. 1, but did not present the calculated MW, which were better to be listed.
4. L. 271, T4 amber27 page à T4 amber27 phage
Author Response
This paper reports on the protein interactions of gp27 of phage T4 with inner-membrane proteins, DamX, SdhA and PpiD, which appear to be important for the DNA intrusion to the host bacterium. They focused on gp27, because the trimeric protein is thought to form a cylinder to facilitate phage DNA intrusion. They not only showed that the host proteins interact with gp27, but also demonstrated that the absence of the host proteins each significantly lower the infection rate of phage T4.
The paper is well organized and easy to read. The findings will be important for the understanding the mechanism of T4 DNA intrusion and important step towards full understanding of phage infection. It will be interesting to general readers. The only thing is that they did not describe their future direction of the research, which the readers may want to see. The followings are minor points which may improve the manuscript.
1. The authors claim that the liposome they used was unilameller. They should refer to any paper which experimentally showed that the liposome they made was unilameller.
Answer: a reference was added (line 222).
2. As the gp27 structure is known, they could envisage the effect of the His-tags’ insertion to gp27-N and gp27-C, which they did not describe.
Answer: Thank you for mentioning this. Yes, we have now added a paragraph discussing this aspect (lines 356-361).
3. The authors presented the apparent MW from the experiment in the legend of Fig. 1, but did not present the calculated MW, which were better to be listed.
Answer: Indeed, there was an error in the terminology. We have now corrected this.
4. L. 271, T4 amber27 page à T4 amber27 phage
Answer: Done, thank you
Round 2
Reviewer 1 Report
Comments and Suggestions for Authors
The authors’ responses are appreciated. However, there are two significant concerns that must be addressed.
i) There is no possible way that the plaques appearing on BL21(DE3) lawn can be due to complemented virus progeny. Complemented viruses will have his-tagged gp27 in the phage structure but the genome would be 27am. These viruses cannot form plaques on suppressor-negative E. coli BL21(DE3), only true recombinants can. Showing that these plaques are due to complementation would be simply wrong, hence this figure must be deleted and appropriate changes made to the text.
On the other hand, complementation is clearly evident in the next experiment and the data shown in Figure 6.
ii) The caveats of assigning certain molecular weight bands to certain cross-linked products, as stated in my original review, must be clearly stated in the main text so that the readers are aware of the potentially soft conclusions made.
